# Effects of a Multifaceted Intervention Program on the Eating Ability of Nursing Home Residents

**DOI:** 10.3390/ijerph18178951

**Published:** 2021-08-25

**Authors:** Mei-Ling Chen, Chia-Hui Chiu, Yueh-Juen Hwu, Shu-Chen Kuo

**Affiliations:** 1Chief of Long-Term Care Department, Jen-Ai Hospital, Taichung 41265, Taiwan; jah0213@mail.jah.org.tw; 2Head Nurse of Nursing Home, Jen-Ai Hospital, Taichung 41265, Taiwan; jahb2@mail.jah.org.tw; 3College of Nursing, Central Taiwan University of Science and Technology, Taichung 40601, Taiwan; yjhwu@ctust.edu.tw

**Keywords:** tongue strength, oral health, nursing home residents, eating ability

## Abstract

Oropharyngeal dysphagia is a common problem for nursing home residents that leads to aspiration pneumonia and malnutrition. Musculature surrounding head and neck and tongue strength are crucial for safe and efficient oropharyngeal swallowing. Oral hygiene facilitates the smooth swallowing. The aim of this study was thus to investigate the effects of a multifaceted intervention program which combines the interactive oral activities, tongue strength training and oral cleaning procedure on the eating ability of nursing home residents. A sequential, multiple time series, single-group quasi-experimental research design was used, and 41 residents were recruited to participate in this study. The study was divided into three phases, each lasting two months. In phase 1, the participants were conducting their usual activities, except that the outcome variables were measured on five occasions to obtain baseline data. Phase 2 was a two-month washout period, and phase 3 was the intervention period. Thirty-six residents completed phase 1, and 34 residents completed the phase 3 intervention period. The eight-week phase 3 intervention program comprised three sessions per day, conducted seven days a week, with each session lasting 15–20 min. Outcome variables were measured at weeks 0, 2, 4, 6, and 8 in phases 1 and 3 to evaluate the feasibility of the program. Following the intervention program, controlling for baseline differences, the patients’ tongue strength, food consumption, mealtime duration, oral health, and dysphagia severity were significantly improved in phase 3 relative to phase 1. These improvements lasted through to at least 2 months posttest. The study illustrates that this multifaceted intervention program may be an effective approach for improving the eating ability of nursing home residents.

## 1. Introduction

Maintaining the eating ability of the residents of long-term care facilities is a common problem. Studies have shown that 54% of people living in residential long-term care facilities have insufficient food intake, and 51% have insufficient fluid intake [1]. A few studies have indicated that the daily calorie intake of residents of long-term care facilities is only 1500 calories or less. Seventy percent of residents’ intake of various nutrients is less than the recommended daily value [2]. Poor eating ability often results in weight loss. Unintentional weight loss is a common problem in nursing home residents, and is associated with adverse, costly clinical outcomes including increases in hospitalization, morbidity, and mortality [1,3]. The need to improve the eating ability of nursing home residents is well recognized universally.

Eating ability is closely related to swallowing function and oral health [4,5]. Natural aging affects the human head and neck anatomy, which in turn can damage the physiological and neural mechanisms that support the swallowing function [6]. Aging-related effects on swallowing may cause dysphagia in the elderly, which is generally considered one of the symptoms of geriatric syndrome. The other symptoms of geriatric syndrome were dementia, depression, and delirium [7]. The prevalence of dysphagia in the elderly varies by study, participants, and the location of the survey [8]. For example, 11–60% of healthy elderly people in the community have chewing and swallowing disorders, while as many as 60–80% of nursing home residents may experience these problems [9]. Dysphagia is a key factor causing malnutrition and aspiration pneumonia in the elderly [10].

During the oral and pharyngeal phases of swallowing, the role of the tongue is to push the food bolus to the teeth for chewing, and then to gather it in the middle of the tongue and push it against the hard palate to force the bolus from the oral cavity into the oropharynx, pharynx, and esophagus. If tongue function is impaired, the ability to manipulate the bolus for chewing, gathering it, and transporting it to the upper digestive tract is affected [11]. Therefore, in the past 10 years, dysphasia-related rehabilitation programs have focused on intensive training of the tongue muscles [12]. The strength of the tongue to lift the bolus and push against the hard palate is critical to swallowing [13]. Many studies have confirmed that intensive training of the tongue muscle can enhance tongue muscle strength [14,15,16], preventing the effects of aging on swallowing function and improving the symptoms of dysphagia [17,18].

Oral health is an indicator of oral muscle condition and chewing ability, both of which affect the ability to eat. Good oral hygiene helps to maintain oral health and chewing ability in the elderly and to reduce their risk of suffering from malnutrition or weight loss [19]. Carrying out oral cleaning procedures has a stimulating effect on the oral cavity and surrounding neuromuscular groups and can ultimately improve swallowing ability [20]. Oral lubrication affects the handling of friction in any two planes of the oral environment, such as that between two teeth, between the teeth and the palate, or between the tongue and the mucous membranes. Generally, lubrication is achieved by the saliva in the mouth, foreign food particles, intraoral drugs, mucosal coating agents, and oral care products. Oral lubrication plays a key role in the effective exercising of human oral tissues, eating, swallowing, speech, and tactile perception functions [21]. Oral cleaning aims to keep the oral cavity clean and lubricated to facilitate smooth swallowing. In addition, interactive oral activities can help to strengthen the swallowing muscles in the elderly, thereby improving their eating ability [22,23].

Once the elderly have the ability to eat, they can increase their nutrient intake and maintain their body weight and a good nutritional status, thus improving their cognitive function and delaying the progression of debilitating conditions [4,5,19,24]. Given the fact that eating ability is related to the musculature surrounding head and neck, tongue strength and oral health, a multifaceted intervention program was developed including interactive oral activities, tongue strength training and oral cleaning procedure.

The aim of this study was to determine the feasibility and efficacy of a multifaceted intervention program for elderly people living in a nursing home. We hypothesized that these effects would improve the participants’ eating ability.

## 2. Methods

### 2.1. Design

The use of rigorous experimental approaches is increasingly encouraged in the social sciences. A true experimental approach is a potent avenue for investigating measures of causality. However, in the nursing home setting, it is difficult to achieve true experimental conditions because of challenges such as small sample sizes and relatively high rates of attrition. We therefore adopted an alternative quasi-experimental approach in this study, and used a sequential, multiple time series, single-group design. To further understand the participants’ subjective experience of the multifaceted intervention program we implemented, one of the researchers conducted an in-depth interview with one-tenth of the participants. One open-ended question was asked in this interview: Can you please tell me how the multifaceted intervention program has affected your daily life?

### 2.2. Sample and Setting

This study was conducted in a nursing home in central Taiwan between October 2020 and April 2021. The inclusion criteria for participants in the study were an age of 65 years or older, residency in the nursing home for at least one month, not medically unstable or in palliative care. In addition, the participants had to be able to understand and conduct the interactive oral activities, tongue strength exercises, and oral cleaning procedure. A total of 41 residents who met the inclusion criteria were enrolled. The exclusion criteria were an age of less than 65 years, absence of signed consent to participate in the study, residency in the nursing home for less than a month, an acute illness or problem within the previous month (such as trauma or infection), and severe cognitive impairment that would have prevented participation in the program. All residents of the nursing home who met the inclusion criteria were recruited for the study. To explore the associations between intervention response and participant characteristics with a phase-level correlation of 0.5 (α = 0.05, β = 0.20), we needed 28 participants. Assuming a loss of approximately 20% during the study, we needed to recruit 34 participants [25].

### 2.3. Multifaceted Intervention Program

All participants received the same intervention protocol, which was based on guidelines for eating and the Taiwanese dysphagia care and guidance protocol [26], a tongue strength-training program [14], and an eating-ability promotion program [23]. There were three aspects covered by this intervention program.

#### 2.3.1. Interactive Oral Activities

To increase the participants’ eating ability, it was necessary to strengthen the muscle groups associated with chewing and swallowing. The interactive oral activities were aimed at strengthening their swallowing-related muscles, and included exercises for the cheek, lip, chin, and throat muscles, along with nasal breathing, cough strengthening, and neck-muscle relaxation [26]. The participants were gathered together in the afternoon every day. To increase the willingness of the participants to engage in these activities, they were designed to be fun and easily incorporated into daily life by incorporating them into activities such as games, phrase solitaire, and singing exercises [23]. For example, blowing beans into a hole, lips holding milk sticks, pronunciation exercises, holding a table tennis ball with the neck, Karaoke, etc.

#### 2.3.2. Tongue Strength Training

This protocol was chosen based on evidence from a recent study by Lin et al. [14], in which the experimental group exhibited greater improvement in both anterior and posterior tongue strength. The training was conducted using Iowa Oral Performance Instrument (IOPI) bulbs. When conducting tongue strength training, the bulb can be positioned behind the central incisors or aligned with the first molars, to target the anterior tongue or posterior tongue, respectively. Participants were asked to squeeze the bulb against the hard palate. When the bulb is positioned near the pharynx, it may elicit an uncomfortable feeling or interfere with swallowing. Elderly people may therefore not tolerate posterior placement of the bulb. The training program was modified to accommodate this as follows: open the mouth, hold the IOPI bulb on the tongue (Figure 1), and squeeze the bulb against the roof of the mouth. The detailed training protocol (translated from the original Mandarin) is presented in Table 1.

C.-H.C. implemented the intervention protocol provided three times per day, seven days per week, for eight weeks. The participants’ adherence to the tongue strength training program was tracked by recording how many repetitions of the exercises they carried out per session in diaries.

#### 2.3.3. Oral Cleaning Procedure

The program aimed to help participants develop or maintain good oral hygiene skills. This part of the intervention program covered the importance of oral hygiene, dry mouth and treatment thereof, using correct brushing techniques, using the soft brush and toothpaste to brush the teeth or denture in sequence to avoid missing any spots, timing and frequency of brushing, using dental floss, denture cleaning and treatment, and demonstrations thereof [23]. It also includes the soft tissue cleaning (e.g., gum and tongue brushing).

### 2.4. Measures

The demographic data collected were age, gender, diet, mealtime context, body mass index (BMI), and calf circumference. Data were collected and recorded by one research nurse, who was informed in detail about the study and was trained in how to perform the tests, administer the instruments listed below, and measure calf circumference and tongue strength. The nurse assessed the participants’ cognitive status using a standardized performance-based assessment (Short Portable Mental Status Questionnaire [SPMSQ]), with a score range from 0 (cognitively intact) to 10 (severely impaired or comatose); their performance in activities of daily living using the Barthel index, with a score range from 0 (completely dependent) to 100 (completely independent), and an instrumental activities of daily living (IADL) scale with a score range from 0 (completely independent) to 32 (completely dependent); and their nutritional status using the Mini Nutritional Assessment (MNA), with a score range from 0 (malnourished) to 30 (normal nourishment).

#### 2.4.1. Measurement of the Maximum Isometric Pressure of the Tongue

In this study, tongue strength was measured on five occasions by the same assessor using the IOPI bulb. Currently, the IOPI bulb is the most widely used instrument for measuring tongue muscle strength [27]. It uses an inflatable tongue bulb and a pressure sensor to measure tongue strength, and its readings are displayed in kilopascals (kPa). The instrument was calibrated once a month, as recommended in the IOPI manual, to ensure measurement accuracy.

The tongue bulb was attached to the posterior side of the alveolar ridge to measure the lingual elevation strength. Participants were instructed to push their tongue up against the bulb with maximum effort. Three trials were conducted, and the maximum pressure registered across the three trials was recorded as the tongue strength [14].

#### 2.4.2. Body Weight

The research nurse conducted independent assessments of body weight using a standardized procedure. In this procedure, the participants were weighed in the morning, before breakfast but after incontinence care, while they were still in their night clothes. The scale was calibrated to zero every time. The nurse weighed every participant monthly.

#### 2.4.3. Amount of Food Consumed

Because collecting data for all three meals was not practical, the amount of food consumed was only measured for the midday meal, since more staff members were available to assist at lunchtime. Each participant was assisted by a nursing aide, and the nursing aides were blinded to any information regarding the participant’s tongue strength, body weight, oral health, and dysphagia severity. The nursing aides measured the amount of food and drink consumed by the participants by subtracting the post meal weight of the food tray from the corresponding premeal tray weight.

#### 2.4.4. Mealtime Duration

The mealtime duration was determined by recording the midday meal time at which the participants began their meal and that at which they finished it.

#### 2.4.5. Oral Health

The Oral Health Assessment Tool (OHAT) was used to assess the participants’ oral health. The scale comprises 10 items: lips, tongue, gums, saliva, teeth, dentures, plaques, toothache, oral mucosa, and oral cleaning. The assessment procedure involves inspecting and palpating the lips and tongue; gently pressing the gum tissue and oral mucosa using a tongue depressor and checking saliva secretion; inspecting teeth, dentures, oral cleaning, and plaque accumulation; and inquiring about toothaches. The minimum score for each item was 0, and the maximum was 2. The total score range was 0–20 points, with higher scores indicating poorer oral health [28].

#### 2.4.6. Gugging Swallowing Screen (GUSS) (Indirect Test)

The Gugging Swallowing Screen developed by Trapl et al. [29] consists of both direct and indirect assessments for the evaluation of an individual’s swallowing ability. In this study we used the indirect test, which comprises five items: assessing concentration (maintaining attention for 15 min); voluntary coughing or clearing the throat twice; swallowing saliva; watching for drooling; checking if the voice became hoarse, wet, or gurgling, or if choking occurs after swallowing saliva. One point is assigned to each item if the condition is normal, and zero points are assigned for abnormality. Thus, the total score range is 0–5 points. If the score is less than 5 points, dysphagia should be suspected and further evaluations, such as the water swallowing test, should be performed.

### 2.5. Data Collection

Upon approval by the institutional review board of Jin-Ai Hospital for studies on human subjects, the authors initiated this six-month research protocol. The study was divided into three phases, each lasting two months. In phase 1, the participants were conducting their usual activities, except that the outcome variables were measured on five occasions (at 0, 2, 4, 6, and 8 weeks) to obtain baseline data. Phase 2 was a two-month washout period, and phase 3 was the intervention period. The intervention period comprised the eight-week multifaceted intervention program, which involved three sessions per day, seven days a week, with each session lasting 15–20 min. The outcome variables were measured at weeks 0 (baseline), 2, 4, 6, and 8, to evaluate the efficacy of the training protocol.

To ensure that the intervention protocol adhered to the design, C.-H.C. conducted all the educational sessions for every participant. All measurements except for food intake and mealtime duration were obtained by another trained research nurse. This research nurse observed each participant from the time of the lunch meal-tray delivery until the meal tray was collected by the nurse for two consecutive days. This two-day lunch assessment procedure was conducted at weeks 0, 2, 4, 6, and 8 to document the effects of the intervention on estimated food intake and mealtime duration. This observational protocol was used to estimate inter-observer reliability (between the nursing aides and the research nurse). The inter-observer reliability coefficients were all over 0.85. One researcher interviewed three participants to assess their general feelings toward the intervention program. The entire interview, which lasted 8–10 min, was recorded for further analysis.

### 2.6. Ethical Considerations

The study was reviewed and approved by the research ethics committee of the hospital (Jin-Ai No. 109-69). The ethical considerations were related to the participants’ autonomy, confidentiality, and anonymity during the study period and upon publication of this report. The objectives of the study were explained to all the residents, and they were informed that they were free to participate, to decline participation, or to withdraw from the study at any time. Written informed consent was obtained from those who agreed to be included in the study or from an alternative decision maker acting on their behalf, and assent to participate was continuously evaluated throughout the study period.

### 2.7. Statistical Analysis

The SPSS for Windows 23.0 package was used for statistical analysis. All data were entered into a database and were independently verified by a second person. Chi-squared and independent *t*-tests were used to verify homogeneity between the phase 1 and phase 3 baselines. A separate analysis of covariance (ANCOVA) was conducted by adjusting the baseline values of the outcome measures to assess the interpretability of the main effects of the intervention between the phases. Between-phase differences in the outcomes were analyzed using a general linear modeling analysis and a repeated measures ANCOVA (RANCOVA). The qualitative data analysis comprised transcription of the content of the recorded interviews as original data, and repeated readings of the transcripts to summarize the participants’ feelings.

## 3. Results

### 3.1. Demographics and Baseline Data

Forty-one of the identified eligible residents participated in the study. Of these, five did not complete the study because they became ill or were discharged, and they were removed from the analyses at phase 1 (leaving *n* = 36). A further five participants withdrew from the study because of a change in their condition during the washout period (phase 2) (leaving *n* = 31); however, three participants were re-included in the study when their condition stabilized in phase 3 (final *n* = 34; Figure 2).

The mean age of the 41 participants was 80.14 years. The majority of the participants were female (63.4%), ate a minced diet (61.0%), and ate predominantly independently (92.7%). The mean reported BMI was 21.42 ± 3.22 kg/m^2^, calf circumference was 28.56 ± 3.80 cm, SPMSQ score was 3.73 ± 2.89, Barthel index was 30.26 ± 21.67, IADL score was 23.31 ± 6.74, and MNA score was 18.20 ± 4.72. The average number of comorbidities was 2.10 ± 0.80 diseases and medication intake was 8.22 ± 2.93 drugs. Table 2 summarizes the participants’ characteristics. There were no statistically significant differences in the participants’ demographic variables between phase 1 and phase 3, which indicates homogeneity (Table 2).

With regard to the baseline outcome values for phases 1 and 3, there were no significant differences in food consumed, mealtime duration, dysphagia severity, or oral health, but tongue strength and body weight did differ significantly (Table 3).

### 3.2. Effects of Multifaceted Intervention Program on Outcome Variables

After ANCOVA is used to control for pretest differences between the two phases, the biweekly measurements show that in phase 3, the participants had significantly improved tongue strength, food consumption, and oral health, as well as shorter mealtimes and less severe dysphagia than in phase 1 (Table 4). As for the OHAT score, improvements were seen between baseline and week 8 assessments for oral mucosa (0.30 ± 0.46 vs. 0.00 ± 0.00, *p* = 0.001), oral cleaning (0.27 ± 0.44 vs. 0.00 ± 0.00, *p* = 0.002), saliva (0.21 ± 0.41 vs. 0.00 ± 0.00, *p* = 0.006), and tongue (0.15 ± 0.35 vs. 0.00 ± 0.00, *p* = 0.023).

### 3.3. Lasting Effects of the Multifaceted Intervention Program

General linear modeling and a sphericity test were used to analyze the correlations between changes in outcome variables over time. RANCOVA with the least significant difference test was used to examine differences in outcome variables across the five time points. This analysis confirmed that there was a significant phase effect on tongue strength (*F* = 41.93, *p* = 0.000), food consumption (*F* = 5.41, *p* = 0.023), oral health (*F* = 7.56, *p* = 0.008), and dysphagia severity (*F* = 5.54, *p* = 0.022). This implies that these outcome variables had improved in phase 3 relative to phase 1.

In terms of the time effect, the RANCOVA revealed a statistically significant improvement in tongue strength (*F* = 53.89, *p* = 0.000), food consumption (*F* = 5.60, *p* = 0.000), mealtime duration (*F* = 5.82, *p* = 0.001), oral health (*F* = 12.64, *p* = 0.000), and dysphagia severity (*F* = 30.08, *p* = 0.000). This implies that tongue strength, food consumption, and GUSS score increased significantly and mealtime duration and oral health score decreased significantly over time.

The time × phase interaction in Table 5 indicates that there was a significant interaction effect between time and phase on tongue strength (*F* = 2.95, *p* = 0.037), mealtime duration (*F* = 3.04, *p* = 0.027), oral health (*F* = 13.21, *p* = 0.000), and dysphagia severity (*F* = 4.70, *p* = 0.005). In other words, there is a difference between the phases in terms of the time effect on these parameters. The RANCOVA did not detect significant main effects of phase, time, or the time × phase interaction on body weight. Pairwise comparisons show that phase 3 of the intervention program resulted in a lasting improvement in tongue strength, food consumption, mealtime duration, oral health, and dysphagia severity (Table 5).

### 3.4. Subjective Feelings of the Participants of the Multifaceted Intervention Program

In-depth interview with one-tenth of the participants was conducted by convenient sampling. Three participants agreed to an in-depth interview. The participants reported that the multifaceted intervention program improved their swallowing function and kept their oral cavity clean. The subjective feelings of the participants were described in Table 6.

## 4. Discussion

The male–female ratio of this nursing home is 4:6. The dropout rates of male participants in the phase 1 and phase 3 were 20% (*n* = 3) and 26% (*n* = 4), respectively. The male–female ratio of the subjects analyzed was not even. The male participants were withdrawn from the study due to a change in condition or hospitalization, or discharge from nursing home. For residents of nursing homes, a BMI of less than 20 kg/m^2^ is indicative of undernutrition [30]. At the beginning of the study, the BMI of the participants was 21.42 ± 3.22 kg/m^2^, indicating that they were not suffering from undernutrition. Calf circumference is one of the indicators of sarcopenia, and a measurement of >34 cm and >32 cm is considered normal for men and women, respectively. In this study, the average calf circumference was 28.56 ± 3.80 cm, which is considered to indicate mild sarcopenia. The average SPMSQ score of 3.73 ± 2.89 is indicative of moderate cognitive impairment, but the participants were still able to follow the instructions given in the intervention program. The Barthel index and IADL score indicate that the participants had a severe level of dependence. They were wheelchair users, but they could still do the tongue-to-palate exercises and interactive oral activities. With assistance, the participants could hold a toothbrush in their hand, or else the care attendants performed their oral cleaning for them. Most of the participants had been diagnosed with at least two diseases, most of which were related to the nervous system (e.g., stroke) or were metabolic diseases (e.g., diabetes). They took more than eight pills per day. Their average nutritional status, 18.20 ± 4.72, indicates that they were at risk of malnutrition. When eating, 92.7% of the participants were able to eat independently in a wheelchair or chair, but 60% of them were on a minced diet.

A pretest–posttest control group design may involve conducting a simple repeated-measures *t*-test to determine whether a treatment has had an impact on the improvement of the outcome. This statistical approach is methodologically somewhat simplistic in nature. Additionally, a randomized controlled study using a quantitative research approach and a multiple time series design has one major drawback: the issue of attrition and the potentially extensive resources that may be needed. The number of residents in the nursing home was not sufficient to provide an adequate sample size to implement a two-group sequential, multiple time series design in this study.

The alternative we used—a sequential, multiple time series design—was thus advantageous, since it was more rigorous and provided the basis for analysis of the longitudinal trajectories of the variables. Conducting data collection on multiple occasions, in particular on more than three occasions, allowed us to identify the longitudinal trajectories of and improvement in variables [31]. This study is the first controlled intervention trial to evaluate the effects of a multifaceted intervention on tongue strength, body weight, food consumption, meal duration, oral health, and dysphagia severity in nursing home residents. Our results showed that the intervention, which involved conducting three sessions per day, seven days a week, for eight weeks, had a significant effect on all outcome measures except body weight.

Tongue strength is crucial for safe and efficient oropharyngeal swallowing. Age-related change in tongue strength puts older adults at risk for difficulty with swallowing [32]. The intervention program in this study, specifically the tongue strength training program and the interactive oral activities, focused on improving oropharyngeal muscle strength, which directly activates the neurophysiological mechanism of swallowing [17]. The primary finding of this study is that Chinese nursing home residents are able to improve their tongue strength during an eight-week multifaceted intervention program.

The results show that the participants’ tongue strength in the intervention phase (phase 3) was higher than in the control phase (phase 1). The measures recorded at weeks 2, 4, 6, and 8 of the intervention phase to explore the time effects on the score changes between the two phases supported the significant phase effects. Table 5 shows that there were significant longitudinal effects on tongue strength. These findings suggest that a period of reinforcement is needed to increase tongue strength. The significant effect of the time × phase interaction on tongue strength indicated that the intervention program achieved a lasting improvement in tongue strength in phase 3. The same was seen in the outcome indicators of improving oral health and dysphagia severity.

The findings of this study indicate that the oral health and dysphagia severity of the participants improved after they had undergone the multifaceted intervention program. Performing oral cleaning procedures thoroughly after each meal would have helped to maintain and improve the participants’ oral health and their chewing and swallowing ability, thereby increasing their ability to eat [33,34,35].

To our knowledge, this nonrandomized, quasi-experimental study with repeated measures and a longitudinal design was the first to examine the potential effects of a multifaceted intervention program on these outcome indicators (tongue strength, body weight, food consumption, mealtime duration, dysphagia severity, and oral health) in Chinese residents of a nursing home. The immediate effects of the improved eating ability were on food consumption and mealtime duration. The long-term effects of the intervention were on the participants’ tongue strength, body weight, dysphagia severity, and oral health. The most consistently documented and statistically significant improvements were with regard to the participants’ tongue strength, mealtime duration, oral health, and dysphagia severity. In contrast, food consumption increased significantly relative to baseline at only some of the four time points. For example, food consumption at the week 4 time point was significantly different from that recorded at weeks 0 and 1. The participants’ body weight, however, did not change over the duration of the intervention.

In terms of body weight, the week 0 measurements for phase 1 and phase 3 were 53.24 ± 9.91 kg and 52.35 ± 10.45 kg, respectively. A clinically significant weight loss episode for nursing home residents is a loss of 5% or more within a 30-day period [30]. The four subsequent body weight measurements for both phase 1 and phase 3 did not reveal this phenomenon, and in fact body weight improved significantly and continuously throughout phase 3 (Table 4). There was no significant time × phase interaction effect on body weight, however, which may be because the participants started the program within the range of normal body weight.

We assessed the participants’ subjective feelings about the intervention program based on interviews with three of the participants, and found that they provided a positive evaluation of the intervention protocol. The qualitative results from the interviews presented here are essential and complementary to the quantitative methods, and are therefore an important aspect of the current research.

There were significant differences in the participants’ baseline values for tongue strength and body weight between phase 1 and 3. The mean tongue strength at the beginning of phase 3 was greater than at the beginning of phase 1, while the mean body weight at the beginning of phase 1 was greater than that at the beginning of phase 3 (Table 3). After adjusting for covariates, the tongue muscle strength recorded during phase 3 was greater than that recorded during phase 1. The effect on body weight was not observed until the fourth week, however. In addition, in phase 1, the baseline tongue muscle strength was 10.31 ± 4.27 kPa. The tongue muscle strength gradually increased during that phase, and had even increased after the two-month washout period. At the beginning of phase 3, the baseline tongue muscle strength was 21.14 ± 7.53 kPa. The ANCOVA was used to compare the values by controlling for the effect of baseline measurements on the later measurements. Table 4 shows that the adjusted biweekly values recorded during phase 3 were consistently higher than those recorded during phase 1, and that this difference was statistically significant. Based on the results of this study, the mean differences of tongue strength between phase 3 and phase 1 may originate from the effects of tongue strength training and interactive oral activities. In addition, we may speculate that the periodic measurement of tongue muscle strength during phase 1 may have prompted the participants to perform tongue-to-palate muscle strengthening exercises, so that the effect of tongue muscle strengthening was present even after the two months of the washout period.

## 5. Limitations

This study has a few important limitations. First, it was conducted in only one nursing home, located in one geographic region. The presence of fewer males than females may elicit selection bias. Thus, these findings may not be generalizable to nursing homes in other geographic regions. In addition, this study lacked a control group and random treatment allocation, and some participants were re-included. Finally, since nursing aides were responsible for measuring the food consumption and mealtime duration, the blinding and validity of these outcome measurements may be limited. However, the sequential, multiple time series, single-group design ensured the homogeneity of the data. Furthermore, C.-H.C. conducted the interventions was the head nurse of the nursing home, and was able to maintain these interventions with sufficient consistency in terms of daily care practice to have consistent effects on participants’ outcome indicators.

## 6. Conclusions and Implications

This study indicates that the multifaceted intervention program introduced here may be an effective method to improve tongue strength, food consumption, mealtime duration, oral health, and dysphagia severity in nursing home residents. Moreover, it proved to be a viable approach to maintaining the participants’ eating ability. This program has been designed to allow the interventions to be incorporated into residents’ daily activities. It should be further evaluated in a wider range of trials to improve the quality of life of nursing home residents.

## Figures and Tables

**Figure 1 ijerph-18-08951-f001:**
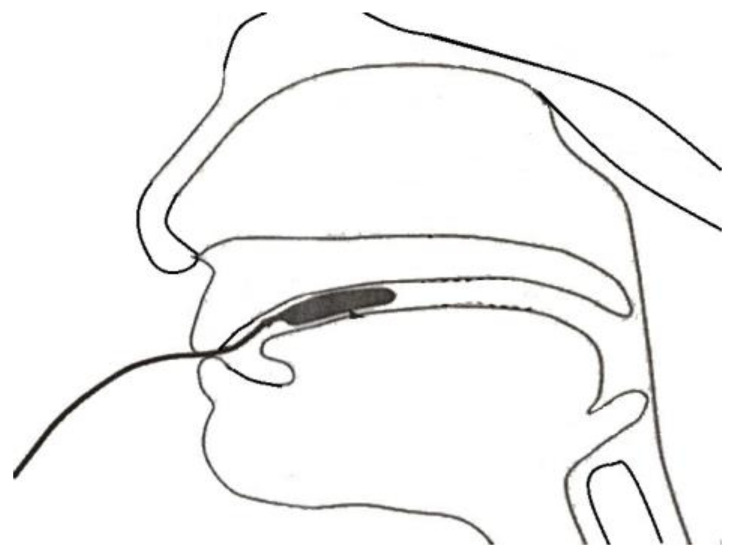
The placement of tongue bulb on tongue.

**Figure 2 ijerph-18-08951-f002:**
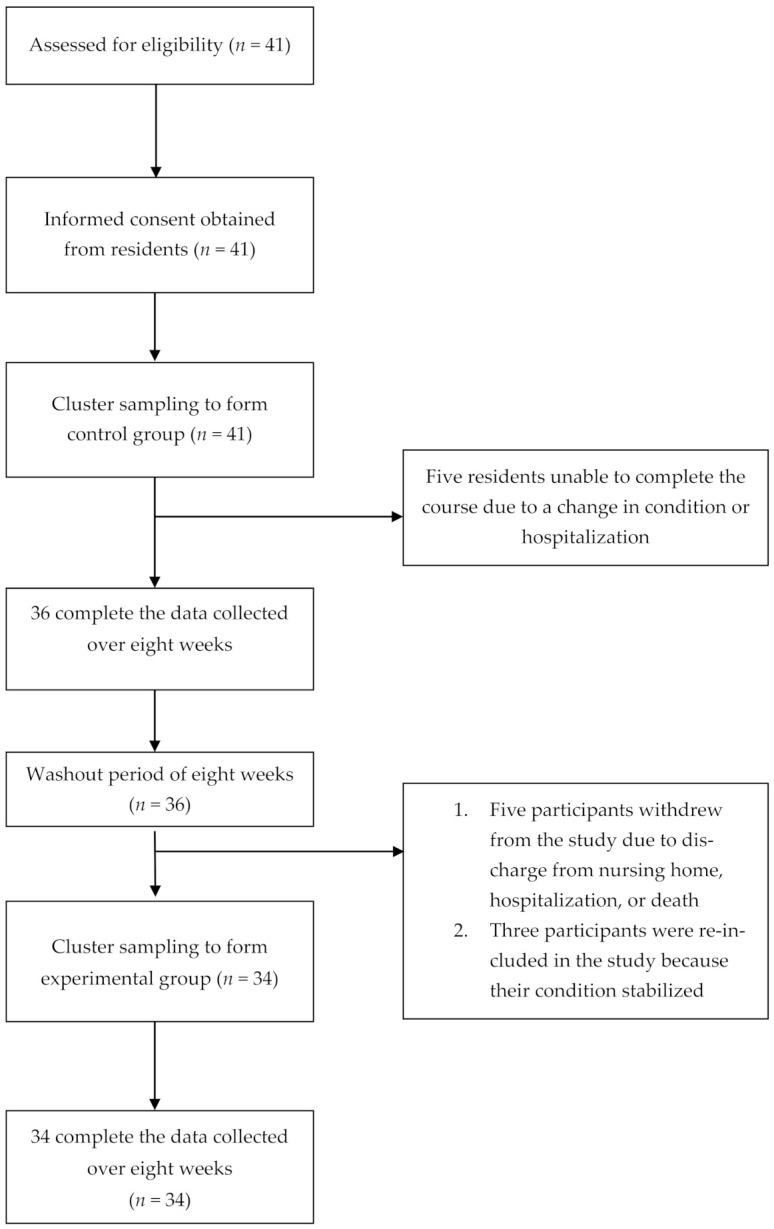
Flow diagram of study participant recruitment.

**Table 1 ijerph-18-08951-t001:** Training protocol step table.

The Training Protocol.	
1	Ask the participant to sit in a chair with a backrest and relax.
2	Instruct them to place the tip of their tongue behind their upper front teeth and feel the edge of the teeth. Tell them that this is where their tongue must go when they swallow, so that the food and liquid are sent back to the pharynx.
3	Instruct the participant to try and place the tip of their tongue behind their upper teeth, and then push it up forcefully. Explain that during the oral phase of swallowing, the tongue moves upward and forward.
4	Put the pressure bulb in the participant’s mouth and lie it flat on the tongue. The seal of the pressure bulb should be behind the front teeth. Once the bulb is in position, mark the connecting tube near the lips with an oil-based pen.
5	Hold the connecting tube with your thumb and index finger and ask the participant to lift their tongue and push the pressure bulb up. Count to six (two seconds). Rest for 5–10 s and repeat the above actions a total of 30 times.
6	Instruct the participant to do the tongue-to-palate pressure bulb exercise 30 min before meals, and then rest for a while before eating. They should perform this exercise three times a day for eight weeks.
7	If the connecting tube shifts, ask the participant to open their mouth, adjust the position of the pressure bulb, and then perform the tongue-to-palate pressure bulb exercise.
8	Verbally encourage the participant, telling them that they have done a good job.
9	Adjust the frequency and timing according to the participant’s condition.
10	Record the number and duration of the tongue-to-palate exercises the participant performs each day in a datasheet.

**Table 2 ijerph-18-08951-t002:** Participant characteristics.

Variable	Total (*n* = 41) n (%)/Mean (SD)	Phase 1 (*n* = 36) n (%)/Mean (SD)	Phase 3 (*n* = 34) n (%)/Mean (SD)	*t*/χ^2^	*p*
Gender				0.364	0.433
Male	15 (36.6)	12 (33.3)	8 (23.5)		
Female	26 (63.4)	24 (66.7)	26 (76.5)		
Diet				8.10	0.088
1. Regular	7 (17.1)	6 (16.7)	15 (44.1)		
2. Soft	6 (14.6)	6 (16.7)	2 (5.9)		
3. Thick mince	12 (29.3)	11 (30.6)	5 (14.7)		
4. Thin mince	13 (31.7)	12 (33.3)	1 (32.4)		
5. Pureed	3 (7.3)	1 (2.8)	1 (2.9)		
Mealtime context				0.00 ^a^	0.967
Independent	38 (92.7)	35 (97.2)	33 (97.1)		
2. Needs help	2 (4.9)	1 (2.8)	1 (2.9)		
3. Feeding	1 (2.4)				
Age	80.14 (10.16)	79.61 (10.62)	79.91 (9.51)	0.124	0.901
Body mass index	21.42 (3.22)	21.80 (3.03)	21.44 (3.11)	−0.487	0.628
Calf circumference	28.56 (3.80)	28.97 (3.54)	28.79 (3.58)	−0.209	0.835
Cognition status (SPMSQ)	3.73 (2.89)	3.56 (2.61)	3.05 (2.25)	−0.849	0.399
Barthel index	30.26 (21.67)	31.69 (21.47)	33.08 (21.81)	0.269	0.789
IADL	23.31 (6.74)	23.05 (6.74)	23.85 (6.72)	0.495	0.622
Number of medications	8.22 (2.93)	8.16 (3.12)	8.47 (3.84)	0.364	0.717
Comorbidities	2.10 (0.80)	2.11 (0.82)	2.12 (0.84)	0.033	0.974
Nutrition status (MNA score)	18.20 (4.72)	19.05 (4.04)	19.23 (4.40)	0.178	0.859

Note: ^a^ Fisher’s exact test.

**Table 3 ijerph-18-08951-t003:** Baseline outcomes of the participants.

Variable	Total (*n* = 41) Mean (SD)	Phase 1 (*n* = 36) Mean (SD)	Phase 3 (*n* = 34) Mean (SD)	*t*	*p*
Tongue strength	9.83 (4.35)	10.31 (4.27)	21.14 (7.53)	7.46	0.000
Body weight (kg)	52.34 (9.95)	53.24 (9.91)	52.35 (10.45)	7.35	0.000
Food consumption (g)	427.07 (241.12)	417.97 (246.17)	437.73 (246.17)	0.34	0.738
Mealtime duration (min)	20.46 (7.39)	20.11 (7.29)	19.26 (4.42)	−0.58	0.562
Dysphagia severity (GUSS score)	4.17 (0.99)	4.19 (0.88)	4.23 (1.04)	0.18	0.860
Oral health (OHAT score)	3.83 (1.94)	3.72 (1.87)	3.38 (2.29)	−0.68	0.499

**Table 4 ijerph-18-08951-t004:** Differences in outcome variables between phases 1 and 3.

Variable	Baseline		Biweekly Measurements		Adjusted Biweekly Measurements		*F*	*p*
	Phase 3 Mean ± SD	Phase 1 Mean ± SD	Phase 3 Mean ± SD	Phase 1 Mean ± SD	Phase 3 Mean ± SD	Phase 1 Mean ± SD		
Tongue strength	21.14 ± 7.53	10.31 ± 4.27						
Week 2			27.73 ± 12.84	14.91 ± 6.11	22.94	19.71	5.25	0.000
Week 4			31.14 ± 12.25	20.05 ± 9.43	26.89	24.31	3.99	0.000
Week 6			33.17 ± 14.66	17.94 ± 7.30	29.16	21.95	3.48	0.001
Week 8			36.38 ± 12.20	20.50 ± 7.62	32.67	24.20	3.69	0.000
Body weight	52.35 ± 10.45	53.24 ± 9.91						
Week 2			52.07 ± 10.53	53.50 ± 9.62	52.51	53.06	57.51	0.000
Week 4			52.75 ± 10.74	53.33 ± 10.44	53.21	52.88	41.09	0.000
Week 6			52.68 ± 10.98	52.87 ± 10.36	53.14	52.41	40.51	0.000
Week 8			52.92 ± 11.05	53.04 ± 10.43	53.38	52.59	37.22	0.000
Food consumption (g)	437.73 ± 246.17	417.97 ± 246.17						
Week 2			483.58 ± 257.95	385.18 ± 237.53	477.19	391.58	6.89	0.000
Week 4			593.64 ± 265.76	440.49 ± 134.64	590.87	443.27	2.87	0.005
Week 6			509.23 ± 261.63	419.47 ± 240.59	503.61	425.10	5.51	0.000
Week 8			628.94 ± 305.25	462.54 ± 279.01	623.52	467.95	4.27	0.000
Mealtime duration (min)	19.26 ± 4.42	20.11 ± 7.29						
Week 2			20.17 ± 5.63	23.58 ± 8.18	20.35	23.40	3.11	0.003
Week 4			20.32 ± 5.77	24.41 ± 6.21	20.53	24.20	4.83	0.000
Week 6			20.23 ± 5.76	20.11 ± 5.52	20.40	19.93	3.99	0.000
Week 8			19.32 ± 5.62	19.88 ± 4.99	19.46	19.74	3.44	0.001
Oral health (OHAT score)	3.38 ± 2.29	3.72 ± 1.87						
Week 2			2.64 ± 1.84	3.75 ± 1.87	2.78	3.60	21.73	0.000
Week 4			2.38 ± 1.45	3.69 ± 1.87	2.50	3.57	16.13	0.000
Week 6			2.29 ± 1.16	3.63 ± 1.89	2.40	3.52	14.20	0.000
Week 8			2.29 ± 1.16	3.86 ± 2.05	2.40	3.74	12.54	0.000
Dysphagia severity (GUSS score)	4.23 ± 1.04	4.19 ± 0.88						
Week 2			4.47 ± 1.07	4.16 ± 0.97	4.45	4.18	11.54	0.000
Week 4			4.82 ± 0.62	4.17 ± 0.97	4.81	4.17	8.62	0.000
Week 6			4.79 ± 0.72	4.16 ± 0.97	4.78	4.18	7.92	0.000
Week 8			3.94 ± 0.23	3.47 ± 0.77	3.94	3.47	3.83	0.000

Note: GUSS, Gugging Swallowing Screen; OHAT, Oral Health Assessment Tool; SD, standard deviation. Adjusted biweekly measurements: Biweekly data after ANCOVA had been used to control for baseline differences between the two phases. Higher scores indicate better tongue strength, greater body weight, greater food consumption, less severe dysphagia, longer mealtime duration, and poorer oral health.

**Table 5 ijerph-18-08951-t005:** Effects of multifaceted intervention program.

Variable	Sphericity Test (*p*)	Mean Squares	Degrees of Freedom	*F*	*p*	LSD Test ^a^
1. Tongue strength	0.000					
Phase		15,171.76	1	41.93	0.000	
Week (0, 2, 4, 6, 8)		2433.45	2.82	53.89	0.000	8 > 4 > 2 > 0, 8 > 6 > 2 > 0
Time × phase		133.18	2.82	2.95	0.037	
2. Body weight	0.000					
Phase		35.81	1	0.07	0.798	
Week (0, 2, 4, 6, 8)		1.90	2.29	0.65	0.544	
Time × phase		8.77	2.29	3.00	0.046	
3. Food consumption (g)	0.000					
Phase		973,060.26	1	5.41	0.023	
Week (0, 2, 4, 6, 8)		187,418.02	4	5.60	0.000	8 > 0, 8 > 2
Time × phase		59,576.60	4	1.78	0.133	
4. Mealtime duration (min)	0.000					
Phase		270.06	1	2.63	0.109	
Week (0, 2, 4, 6, 8)		145.52	3.21	5.82	0.001	2 > 8, 4 > 0, 4 > 8
Time × phase		76.18	3.21	3.04	0.027	
5. Oral health (OHAT score)	0.000					
Phase		112.30	1	7.56	0.008	
Week (0, 2, 4, 6, 8)		8.24	1.82	12.64	0.000	0 > 2, 0 > 4, 0 > 6, 0 > 8
Time × phase		8.61	1.82	13.21	0.000	
6. Dysphagia severity (GUSS score)	0.000					
Phase		15.39	1	5.54	0.022	
Week (0, 2, 4, 6, 8)		11.24	2.58	30.08	0.000	4 > 0 > 8, 6 > 0 > 8, 2 > 8
Time × phase		1.76	2.58	4.70	0.005	

^a^ LSD, least significant difference; comparisons between two means among the baseline and four subsequent biweekly measurements. 8 > 0 means the mean of the week 8 measurement is significantly greater than that of the week 0 (baseline) measurement, based on the LSD test. Higher scores indicate greater tongue strength, greater body weight, greater food consumption, less severe dysphagia, longer mealtime duration, and poorer oral health.

**Table 6 ijerph-18-08951-t006:** Participants’ feedback on the multifaceted intervention program.

Theme	Statement
1. To improve swallowing function	“In the past, when I had meals, I often coughed and had no confidence. I was also afraid of being looked down upon by others. The thought of having a meal frightened me. When I eat now, I can keep my food moving smoothly without it getting stuck, which leaves me feeling comfortable and vigorous.” “I like chicken very much, but I used to eat only boiled chicken, which didn’t taste good. Now I can even eat chicken thighs, and I don’t bite my tongue. I feel very happy.” When I saw other people’s plates, I felt embarrassed. Now I can eat rice and non-pureed meat and vegetables that are cut into pieces. I feel that I have dignity with respect to eating.”
2. To keep the oral cavity clean	“I have followed the attendant’s reminder to brush my teeth and rinse my mouth after meals. I feel refreshed, and I don’t always need to clear my throat and cough.” “I brush my teeth and rinse my mouth after all three meals. The discomfort in my oral cavity has reduced a lot, and I don’t have to go to the dentist as often.” “Ever since I have known that dental plaques form within three minutes of a meal, and that the plaques are full of bacteria that can induce systemic diseases, I am serious about brushing my teeth for at least three minutes and within three minutes after meals. Now I implement oral hygiene very carefully every day. I feel very busy every day.”

## Data Availability

The datasets used and/or analyzed during the present study are available from the corresponding author on reasonable request.

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
