# Peer review of "Effects of a Multifaceted Intervention Program on the Eating Ability of Nursing Home Residents"

_ijerph, 2021, doi:10.3390/ijerph18178951_

Round 1
Reviewer 1 Report
Effects of a multifaceted intervention program on the eating ability of nursing home residents
This study investigated the effects of a multifaceted intervention program on the eating ability of nursing home residents. Although the topic and results are valuable, there are several points required reconsideration in this manuscript.
Introduction
Please specify the hypothesis of this study.
P2L71
Please clarify the definition of “Multifaceted intervention”. Does it mean training that includes multiple aspects of strength and oral hygiene?
Residents in nursing homes include people with different ADL states. In the introduction description, training is considered to be applicable to all residents. It may be necessary to discuss the target resident to which training is applicable as in this study.
Materials and Methods
P3L112-114
It is difficult to image how to incorporate the participants into activities like games, phrase solitaire, and singing exercises. Were these activities also included as the program?? Please mention the relationship the program and these activities concretely.
P3L118-119
It is better to add the photo of the IOPI bulbs to image the training tool.
P3L124-127
Concerning the training, although the text explains that it is 3 steps, the table describes 10 steps. So, it is difficult to understand the description about the training. You should specify the 3 steps into the table.
P4 L133-137 Oral cleaning procedure
Descriptions about oral cleaning are lacking. Please provide detailed description about correct tooth brushing or denture cleaning method including the tool (brush, toothpaste, denture cleaner etc.) or duration.
Is it include the soft tissue cleaning? (Tongue brushing etc.)
Was the oral cleaning method and required duration the same regardless of the number of remaining teeth? If the cleaning method or process differs depending on the number of remaining teeth, please specify it.
P5L176 Mealtime duration
Did you also measure only for the midday meal? You should state clearly.
Results
Since oral hygiene guidance and cleaning are provided in this study, it is necessary to clarify not only the OHAT score but also the oral hygiene status of the subjects and their changes.
P7 Figure1
The n number in the figure has missed. (Data collected over eight weeks)
P7L246
Did the three participants re-included managed during the withdrew period? Is it appropriate to include three participants who had an unmanaged period for analysis?
P7L304-330
Participant's comments are too long to get the point. Please try to simplify the description by using a table.
Did you interview all the participants? And when did you do the interview? If the interview is included in the survey items, please also mention the interview precisely in the methods section.
P7L325-326
“The discomfort in my oral cavity has reduced a lot, and I don’t have to go to the dentist as often.”
Did participant 2 go to the dentist often because of discomfort despite being a nursing home resident?
Discussion
Please add a consideration about the high dropouts of male participants. Also, please add a discussion about the fact that the male-female ratio of the subjects analyzed was not even.
P12L367-
Please add a discussion about the very low tongue pressure of participants at baseline. It is also considered rare for training to more than double the tongue pressure. Please add a discussion on this point as well.
Limitations
How about explaining the difference in the male-female ratio in this section as well?
Conclusion
The Conclusions and implication section includes trials of cost-effectiveness that are not directly related to this study. These should be explained in the discussion, and this section should include content that is directly linked to the conclusion.
Thank you for giving the opportunity to review this manuscript.
Author Response
Manuscript Reference Number: ijerph-1319289
Title: Effects of a multifaceted intervention program on the eating ability of nursing home residents
Journal: IJERPH
|
Reviewer #1 comments |
Response to reviewer #1 |
|
1. Introduction |
|
|
(1) Please specify the hypothesis of this study. |
(1) We thank Reviewer 1 for the constructive comments. Six hypotheses have added to the section of introduction. “Our research hypotheses are that: (1) Multifaceted intervention program will increase the maximal isometric tongue strength. (2) Multifaceted intervention program will increase the body weight. (3) Multifaceted intervention program will facilitate the food consumption. (4) Multifaceted intervention program will decrease the mealtime duration. (5) Multifaceted intervention program will improve the oral health. (6) Multifaceted intervention program will decrease the dysphagia severity.” (p.2) |
|
(2) P2L71 Please clarify the definition of “Multifaceted intervention”. Does it mean training that includes multiple aspects of strength and oral hygiene? |
(2) We have clarified the definition of multifaceted intervention “Given that fact eating ability is related to the musculature surrounding head and neck, tongue strength and oral health, a multifaceted intervention program was developed. This program includes interactive oral activities, tongue strength training and oral cleaning procedure to improve eating ability in the elderly.” (p.2) |
|
(3) Residents in nursing homes include people with different ADL states. In the introduction description, training is considered to be applicable to all residents. It may be necessary to discuss the target resident to which training is applicable as in this study.
|
(3) The mean ADL was 30.26 ± 21.67. All the participants can sit in the wheelchairs to eat. “The inclusion criteria for participants in the study were an age of 65 years or older, residency in the nursing home for at least one month, not medically unstable or in palliative care, and an ability to participate in a multifaceted intervention program.” (p.3) “The Barthel index and IADL score indicate that the participants had a severe level of dependence. They were wheelchair users, but they could still do the tongue-to-palate exercises and interactive oral activities. With assistance, the participants could hold a toothbrush in their hand, or else the care attendants performed their oral cleaning for them.” (p.14) |
|
2. Materials and Methods |
|
|
(1) P3L112-114 It is difficult to image how to incorporate the participants into activities like games, phrase solitaire, and singing exercises. Were these activities also included as the program?? Please mention the relationship the program and these activities concretely |
(1) We appreciate the important points that Reviewer 1 raised with regard to the methods. We have added the description to the text. “The participants were met together in the afternoon every day. To increase the willingness of the participants to engage in these activities, they were designed to be fun and easily incorporated into daily life by incorporating them into activities like games, phrase solitaire, and singing exercises [23]. For example, blowing beans into a hole, lips holding milk sticks, pronunciation exercises, holding a table tennis ball with the neck, Karaoke etc.” (p.3) |
|
(2) P3L118-119 It is better to add the photo of the IOPI bulbs to image the training tool. |
(2) We add the picture of the IOPI bulbs to the text. (p.4) |
|
(3) P3L124-127 Concerning the training, although the text explains that it is 3 steps, the table describes 10 steps. So, it is difficult to understand the description about the training. You should specify the 3 steps into the table. |
(3) To avoid confusion, the number of three steps was deleted. “The training program was modified to accommodate this as follows: open the mouth, hold the IOPI bulb on the tongue (Figure 1), and squeeze the bulb against the roof of the mouth. The detailed training protocol (translated from the original Mandarin) is presented in Table 1.” (p.3) |
|
(4) P4 L133-137 Oral cleaning procedure Descriptions about oral cleaning are lacking. Please provide detailed description about correct tooth brushing or denture cleaning method including the tool (brush, toothpaste, denture cleaner etc.) or duration. |
(4) Descriptions about oral cleaning are added to the text “The program aimed to help participants develop or maintain good oral hygiene skills. This part of the intervention program covered the importance of oral hygiene, dry mouth and treatment thereof, using correct brushing techniques, using the soft brush and toothpaste to brush the teeth or denture in sequence to avoid missing any spots, timing and frequency of brushing, using dental floss, denture cleaning and treatment, and demonstrations thereof [23]. It also includes the soft tissue cleaning (e.g. gum and tongue brushing).” (p.5) |
|
(5) Is it include the soft tissue cleaning? (Tongue brushing etc.) |
(5) Yes. “It also includes the soft tissue cleaning (e.g. gum and tongue brushing).” (p.5) |
|
(6) Was the oral cleaning method and required duration the same regardless of the number of remaining teeth? If the cleaning method or process differs depending on the number of remaining teeth, please specify it. |
(6) “With assistance, the participants could hold a toothbrush in their hand, or else the care attendants performed their oral cleaning for them” (p.14) The oral cleaning process includes the remaining teeth and denture. |
|
(7) P5L176 Mealtime duration Did you also measure only for the midday meal? You should state clearly. |
(7) This sentence has been revised. “The mealtime duration was determined by recording the midday meal time at which the participants began their meal and that at which they finished it.” (p.6) |
|
3. Results |
|
|
(1) Since oral hygiene guidance and cleaning are provided in this study, it is necessary to clarify not only the OHAT score but also the oral hygiene status of the subjects and their changes. |
(1) We have added the description about the oral hygiene status of the subjects and their changes. “As for the OHAT score, there were significant improvements in the removing plaque and moisture of oral mucous membrane.” (p.11) |
|
(2) P7 Figure1 The n number in the figure has missed. (Data collected over eight weeks) |
(2) This Figure has been revised (p.9). |
|
(3) P7L246 Did the three participants re-included managed during the withdrew period? Is it appropriate to include three participants who had an unmanaged period for analysis? |
(3) The data of this three participants were not in the analysis of phase 1. In addition, we have mention this issue in the section of limitations. “In addition, this study lacked a control group and random treatment allocation, and some participants were re-included. Given these facts, there is lack of power.” (p.16) |
|
(4) P7L304-330 Participant's comments are too long to get the point. Please try to simplify the description by using a table. |
(4) We have revised this paragraph and added one Table (Table 6) “3.4. Subjective feelings of the participants of the multifaceted intervention program Three participants agreed to an in-depth interview. The participants reported that the multifaceted intervention program improved their swallowing function and kept their oral cavity clean. The subjective feelings of the participants were described in Table 6.” (pp.13-14)
|
|
(5) Did you interview all the participants? And when did you do the interview? If the interview is included in the survey items, please also mention the interview precisely in the methods section. |
(5) Three participants agreed to be interviewed after the end of phase 3. “Three participants agreed to an in-depth interview. The participants reported that the multifaceted intervention program improved their swallowing function and kept their oral cavity clean. The subjective feelings of the participants were described in Table 6.” (p.13) |
|
(6) P7L325-326 “The discomfort in my oral cavity has reduced a lot, and I don’t have to go to the dentist as often.” Did participant 2 go to the dentist often because of discomfort despite being a nursing home resident? |
(6) This is an affiliated nursing home of hospital. The hospital is located nearby. The participant can go to the dentist very conveniently. This participant just expressed his subjective feeling. |
|
4. Discussion |
|
|
(1) Please add a consideration about the high dropouts of male participants. Also, please add a discussion about the fact that the male-female ratio of the subjects analyzed was not even. |
(1) This phenomenon is explained in the text “The male-female ratio of this nursing home is 4:6. The dropout rates of male participants in the phase 1 and phase 3 were 20 % and 34 %, respectively. The male-female ratio of the subjects analyzed was not even.” (p.14) |
|
(2) P12L367- Please add a discussion about the very low tongue pressure of participants at baseline. It is also considered rare for training to more than double the tongue pressure. Please add a discussion on this point as well. |
(2) This point has been added to the discussion “The ANCOVA was used to compare the values by controlling for the effect of baseline measurements on the later measurements. Table 4 shows that the adjusted biweekly values recorded during phase 3 were consistently higher than those recorded during phase 1, and that this difference was statistically significant. Based on the results of this study, the mean differences of tongue strength between phase 3 and phase 1 may originate from the effects of tongue strength training and interactive oral activities. In addition, we may speculate that the periodic measurement of tongue muscle strength during phase 1 may have prompted the participants to perform tongue-to-palate muscle strengthening exercises, so that the effect of tongue muscle strengthening was present even after the two months of the washout period.” (p.16) |
|
5. Limitations |
|
|
(1) How about explaining the difference in the male-female ratio in this section as well? |
(1) The male-female ratio is added to the limitations “The presence of fewer males than females may elicit selection bias.” (p.16)
|
|
6. Conclusion |
|
|
(1) The Conclusions and implication section includes trials of cost-effectiveness that are not directly related to this study. These should be explained in the discussion, and this section should include content that is directly linked to the conclusion. |
(1) The conclusion has been revised “This program has been designed to allow the interventions to be incorporated into residents’ daily activities. It should be further evaluated in a wider range of trials to improve the quality of life of nursing home residents.” (p.16) |
Reviewer 2 Report
Effects of a multifaceted intervention program on the eating 2 ability of nursing home residents
Dear authors,
This was a very interesting and useful study. You list your limitations that it may not be generalizable, however, the usefulness outweighs that concern.
Please include the lack of power as one additional limitation.
I have no other major recommendations.
Trivial issues are the following:
Abstract
More detail is needed describing the procedure and in statistical results. The abstract should be a “mini” manuscript.
Introduction
In the U.S., the spelling is calorie
Line 39: Explain geriatric syndrome.
Line 49: Re-write to: …chewing, gathering it, and transporting it…
Table 1 and 3 Column alignment is off.
Author Response
Manuscript Reference Number: ijerph-1319289
Title: Effects of a multifaceted intervention program on the eating ability of nursing home residents
Journal: IJERPH
|
Reviewer #2 comments |
Response to reviewer #2 |
|
1. Please include the lack of power as one additional limitation. |
We thank Reviewer 2 for the constructive comments. This point has been added to the limitation “In addition, this study lacked a control group and random treatment allocation, and some participants were re-included. Given these facts, there is lack of power.” (p.16) |
|
2. Abstract |
The abstract has been revised to be “Oropharyngeal dysphagia is a common problem for nursing home residents that leads to aspiration pneumonia and malnutrition. Musculature surroding head and neck and tongue strength are crucial for safe and efficient oropharyngeal swallowing. Oral hygiene facilitates the smooth swallowing. The aim of this study was thus to investigate the effects of a multifaceted intervention program which combines the interactive oral activities, tongue strength training and oral cleaning procedure on the eating ability of nursing home residents. A sequential, multiple time series, single-group quasi-experimental research design was used, and 41 residents were recruited to participate in this study. The study was divided into three phases, each lasting two months. In phase 1, the participants were conducting their usual activities, except that the out-come variables were measured on five occasions to obtain baseline data. Phase 2 was a two-month washout period, and phase 3 was the intervention period. Thirty-six residents completed phase 1, and thirty-four residents completed the phase 3 intervention period. The eight-week phase 3 intervention program comprised three sessions per day, conducted seven days a week, with each session lasting 15–20 min. Outcome variables were measured at weeks 0, 2, 4, 6, and 8 in phases 1 and 3 to evaluate the feasibility of the program. Following the intervention program, controlling for baseline differences, the patients’ tongue strength, food consumption, mealtime duration, oral health, and dysphagia severity were significantly improved in phase 3 relative to phase 1. These improvements lasted through at least 2-month posttest. The study illustrates that this multifaceted intervention program may be an effective approach for improving the eating ability of nursing home residents.” |
|
3. Introduction |
|
|
(1) In the U.S., the spelling is calorie |
(1) It has been corrected “A few studies have indicated that the daily calorie intake of residents of long-term care facilities is only 1,500 calories or less.” (p.1) |
|
(2) Line 39: Explain geriatric syndrome. |
(2) The geriatric syndrome has been explained: “Aging-related effects on swallowing may cause dysphagia in the elderly, which is generally considered one of the symptoms of geriatric syndrome. The other symptoms of geriatric syndrome were dementia, depression, and delirium [7].” (p.1-2) |
|
(3) Line 49: Re-write to: …chewing, gathering it, and transporting it… |
(3) The sentence has been rewritten “If tongue function is impaired, the ability to manipulate the bolus for chewing, gathering it, and transporting it to the upper digestive tract is affected [11].” (p.2) |
|
4. Table 1 and 3 Column alignment is off. |
These have been corrected. |
Round 2
Reviewer 1 Report
Effects of a multifaceted intervention program on the eating ability of nursing home residents
In this manuscript, many points have been revised. Although many points have been clarified, there are some points that need further correction.
Introduction
Thank you for adding the hypothesis. However, the hypothesis statements are too long with repeating the same words. Please correct it to a concise description.
Materials and Methods
P3L104
You need to specifically state what abilities you have identified and determined that the participant has the "ability to participate in the program" in the inclusion criteria of the method.
Results
P11L297
If you have mentioned plaque and moisture improvements in your results, you need to not only state that there were significant improvements, but also explicitly state the value of that score to indicate the improvement.
P13L333
If you conducted the interview with the participants, please add the method section regarding the interview. Please describe whether the interview was requested to all participants and the results were obtained from three participants, or whether they were requested to a specific participant (if so, the request criteria are also specified).
Discussion
P14L343
Is there anything you can discuss about why the dropout rate of male were higher than that of female?
Thank you for giving the opportunity to review this manuscript. I hope that the authors revise and reply to the stated points and review again.
Author Response
Manuscript Reference Number: ijerph-1319289
Title: Effects of a multifaceted intervention program on the eating ability of nursing home residents
Journal: IJERPH
|
Reviewer #1 comments |
Response to reviewer #1 |
|
1. Introduction |
|
|
(1) Thank you for adding the hypothesis. However, the hypothesis statements are too long with repeating the same words. Please correct it to a concise description. |
(1) We thank Reviewer 1 for the constructive comments. “Given that fact eating ability is related to the musculature surrounding head and neck, tongue strength and oral health, a multifaceted intervention program was developed including interactive oral activities, tongue strength training and oral cleaning procedure. Thus, the aim of this study was to determine the effects of a multifaceted intervention program on eating ability in elderly people living in a nursing home. We hypothesized that the multifaceted intervention program would lead to improvements in tongue strength, body weight, food consumption, oral health, and decreasing the mealtime duration and dysphagia severity as compared to a regular living pattern.” |
|
2. Materials and Methods |
|
|
(1) P3L104 You need to specifically state what abilities you have identified and determined that the participant has the "ability to participate in the program" in the inclusion criteria of the method. |
(1) “In addition, the participants can understand and conduct the interactive oral activities, tongue strength exercises, and oral cleaning procedure.” |
|
3. Results |
|
|
(1) P11L297 If you have mentioned plaque and moisture improvements in your results, you need to not only state that there were significant improvements, but also explicitly state the value of that score to indicate the improvement. |
(1) “As for the OHAT score, improvements were seen during the phase 3, for example the item of removing plaque (from baseline 0.53 ± 0.56 to week 8 0.41 ± 0.50) and moisture of oral mucous membrane (from baseline 0.30 ± 0.46 to week 8 0.00 ± 0.00).” |
|
(2) P13L333 If you conducted the interview with the participants, please add the method section regarding the interview. Please describe whether the interview was requested to all participants and the results were obtained from three participants, or whether they were requested to a specific participant (if so, the request criteria are also specified). |
(2) “To further understand the participants’ subjective experience of the multifaceted intervention program we implemented, one of the researchers conducted an in-depth interview with one-tenth of the participant. One open-ended question was asked in this interview: Can you please tell me how the multifaceted intervention program has affected your daily life? ”
“In-depth interview with one-tenth of the participant was conducted by convenient sampling.” |
|
4. Discussion |
|
|
(1) P14L343 Is there anything you can discuss about why the dropout rate of male were higher than that of female? |
(1) “The male-female ratio of this nursing home is 4:6. The dropout rates of male participants in the phase 1 and phase 3 were 20 % (n = 3) and 26 % (n = 4), respectively. The male-female ratio of the subjects analyzed was not even. The male participants were withdrawn from the study due to a change in condition or hospitalization, or discharge from nursing home.” |
